# The Impact of COVID-19 on the Insurance Industry

**DOI:** 10.3390/ijerph17165766

**Published:** 2020-08-10

**Authors:** Pius Babuna, Xiaohua Yang, Amatus Gyilbag, Doris Abra Awudi, David Ngmenbelle, Dehui Bian

**Affiliations:** 1School of Environment, Beijing Normal University, Beijing 100875, China; 201939180009@mail.bnu.edu.cn (P.B.); 201931180020@mail.bnu.edu.cn (D.B.); 2Department of Geography and Environmental Science, The University of Reading, Whiteknights, P.O. Box 227, Reading RG6 6AB, UK; 3College of Agriculture and Natural Resources, Kwame Nkrumah University of Science and Technology, Kumasi PMB KNUST, Ghana; amatusmike@yahoo.com (A.G.); dorisawudi@outlook.com (D.A.A.); yngmen2018@gmail.com (D.N.); 4Chinese Academy of Agricultural Sciences (CAAS), Institute of Environment and Sustainable Development in Agriculture (GSCAAS), Haidian District, Beijing 100875, China; 5Department of Nutrition and Food Hygiene, School of Public Health, Nanjing Medical University, Longmian Avenue 101, Nanjing 211166, China

**Keywords:** coronavirus, COVID-19, insurance industry, infection rate, National Insurance Commission, pandemic, World Health Organization

## Abstract

This study investigated the impact of COVID-19 on the insurance industry by studying the case of Ghana from March to June 2020. With a parallel comparison to previous pandemics such as SARS-CoV, H1N1 and MERS, we developed outlines for simulating the impact of the pandemic on the insurance industry. The study used qualitative and quantitative interviews to estimate the impact of the pandemic. Presently, the trend is an economic recession with decreasing profits but increasing claims. Due to the cancellation of travels, events and other economic losses, the Ghanaian insurance industry witnessed a loss currently estimated at GH Ȼ112 million. Our comparison and forecast predicts a normalization of economic indicators from January 2021. In the meantime, while the pandemic persists, insurers should adapt to working from remote locations, train and equip staff to work under social distancing regulations, enhance cybersecurity protocols and simplify claims/premium processing using e-payment channels. It will require the collaboration of the Ghana Ministry of Health, Banking Sector, Police Department, Customs Excise and Preventive Service, other relevant Ministries and the international community to bring the pandemic to a stop.

## 1. Introduction

Coronavirus disease, scientifically reclassified as COVID-19, has assumed global pandemic proportions [1]. It attained a pandemic status declared by the World Health Organization (WHO) on 11 March 2019 [1]. The current spread of the virus at a fast rate compared to previous pandemics has resulted in a total lockdown of nations, ban on travels, public gatherings and closure of offices. There has been global closure of businesses as well as the loss of jobs and lives. The general economic situation is a global recession. In most instances, the insurance industry and governments all over the world have become the beacons of hope to which people look for rescue from total annihilation. However, due to the fast increase in infection cases greater than the recovery of infected people, the pandemic has overwhelmed many governments and financially weakened some insurance companies. The impact of the pandemic on the Ghanaian insurance industry is yet to be estimated and projected to provide a guide for government and insurers for the simulation of future events. 

As of 22 June 2020, 14,007 confirmed cases have been reported in Ghana, and the numbers keep increasing by day [1]. On 30 March, the president of Ghana announced a partial lockdown in Accra, Tema and Kumasi. The lockdown, however, lasted for only three weeks and was lifted on 19 April following the overweight of economic meltdown on the nation. Though the country received a financial support of USA $1 billion and donations from other organizations, philanthropists and partners to cushion the impact of the pandemic, the economic effects of the partial lockdown were much greater than can be contained by the nation. Increased pressure on health workers and hospital facilities at present is overbearing, and it seems a catastrophic situation is imminent.

The insurance industry in Ghana plays a very important role in the national economy [2]. Ghana presently has a huge infrastructural deficit that requires the establishment of efficient insurance policies to pursue economic development [3,4]. The industry provides safety and stability to individuals, groups, institutions and businesses [5]. A healthy and developed insurance industry will improve the stability of financial markets. Over a long period, the Ghanaian insurance industry has witnessed stability and economic growth due to the stable growth of the economy boosted by increased cocoa, gold and agricultural production. The recent boom in the oil business has particularly catapulted insurance premiums [2]. Currently, the total assets of the Ghanaian insurance industry is GH Ȼ6.2 billion. The Life sector contributes GH Ȼ3.1 billion, Non-Life GH Ȼ2.4 billion and Reinsurance GH Ȼ0.7 billion. Thirty percent (30%) of Ghanaians (i.e., 9,311,985 of 31,039,949) are currently insured while twelve thousand (12,000) Ghanaians are employed in the insurance industry [2]. In 2018, the total annual profit of the insurance industry was GH Ȼ202 million while total premium was GH Ȼ2.9 billion. The average daily benefit paid by Life insurers is GH Ȼ1.9 million and average daily claim is GH Ȼ1.1 million [2]. Currently, the insurance industry pays a total cooperate tax of GH Ȼ36 million. 

Historically, zoonosis (i.e., the transmission of diseases from animals to humans) has been identified as the major cause of global pandemics [6]. Antonine plague, believed to be Smallpox, is the earliest record of a global pandemic that occurred between 165–180 AD, killing 5 million people, and causing USA $11–25 million financial loss [7]. The Black Death (1347–1351) caused by Yersinia pestis, bacteria found in rats, was a devastating pandemic that killed 200 million people all over the world and caused USA $70–120 million financial loss [8,9]. The Spanish flu occurred from 1918 to 1919 and was caused by an H1N1 virus that lives in pigs [10,11,12]. The flu killed between 40–50 million people and caused an economic loss of USA $70–90 million [13,14]. HIV AIDS has been in existence since 1981 until now [15,16,17]. It is caused by the Human Immunodeficiency Virus that lives on chimpanzees [18]. AIDS has killed between 25–35 million people so far and caused USA $100–200 million in financial loss to the world [19,20]. More recently, the swine flu started in 2009 to 2010 [20,21,22,23]. It is caused by an H1N1 virus in pigs and has killed 200,000 people as well as caused USA $10–20 million global financial loss [24,25]. SARS-CoV started in 2002 and ended in 2003 [26]. It is caused by a coronavirus in bats and civet cats and has killed 770 people while causing USA $1–3 million financial loss to the global economy [27,28]. Ebola occurred in the West African sub-region from 2014 to 2016 [29,30,31,32]. It is caused by the ebolavirus in wild animals [33,34,35,36,37,38,39,40,41,42] and has killed 11,000 people while causing USA $16–30 million financial loss to the global economy. Five years ago, MERS started and is still present. It is caused by a coronavirus from bats and camels and killed 850 people so far [43]. Its total economic loss is USA $2–9 million and still counting [43]. The origin of the novel COVID-19 at present is still unknown, and its total economic loss is still counting.

While there is limited literature on the impact of pandemics on the insurance industry, a parallel comparison can be made to the impact of other natural disasters on the global financial system. Dimitri [44] studied how available economic resources could be shared by health authorities to stop the spread of epidemic diseases. His analysis suggests that to be able to effectively distribute economic resources optimally depends on the cost functions namely: the available technology for controlling the relevant parameters underlying the epidemic and the available financial resources. Lagoarde-Segot and Leoni [45] used a theoretical model to study the stability of the banking sector in the joint prevalence of Malaria and AIDS. Their studies suggest that the likelihood of collapse of the banking sector increases as the prevalence of pandemics increases. Fan et al. [46] estimated national losses in national incomes in the US economy that might occur in the event of a major influenza-related pandemic. Their study also covered the number of lives lost as a result of the pandemic. They further estimated the expected number of influenza-pandemic related deaths to reach 720,000 per year and calculated annual losses as a result of pandemic impacts to reach USA $500 million. Bongini et al. [47] studied the profitability or losses of banks in crises such as a global pandemic. Using a sample of 109 European bank holding companies from 2006 to 2016, they studied the main drivers of profitability shocks and factors that help banks bounced back avoiding a more severe situation. Their findings showed that the crisis has its stocks in the lending activity and in the deterioration loan portfolio caused by a risk that is not counterbalanced by adequate loan loss provisions and capitalizations. They also discovered that banks that recovered after the shock adopted a more conservative lending policy, reduced their rates and performed better afterward. Sigala [48] reviewed literature to understand the impacts of COVID-19 on the tourism industry. Her study revealed that COVID-19 has different impacts on the tourism industry, which are based on the characteristic nature of the particular tourism sector, its size, location, management and ownership style. She further noted that the highly heterogeneous nature of tourism demand also determines COVID-19 impacts and implications. They concluded that COVID-19 tourism research should not only discuss the different impacts of COVID-19 on tourism but also provide an explanation of the roots of such differences and the scope to test any suggestions on how to address possible inequalities that may arise from impacts of COVID-19. Barro et al. [49] studied the potential effect of the coronavirus on mortality and economic activity by comparing lessons from the 1918–1920 Great Influenza Pandemic. With a parallel evaluation of economic declines in GDP, increased consumption, deaths, decreased realized real returns on stocks and especially on short-term government bills caused by the 1918–1920 Great Influenza Pandemic, the study compared these economic impacts to that caused by the Coronavirus in our current dispensation to draw lessons from previous experiences. The study concluded that the current situation decline in stock prices, increases in stock-price volatility, decreases in nominal interest rates and contractions of real economic activity are familiar outcomes that require quick responses. Cerra and Sexena [50] investigated the behavior of output following financial and political crises in 190 countries. They particularly examined the economic impact of deterioration in a country’s political governance. Alfaro et al. [51] investigated aggregate and firm level stock returns during pandemics in real time. Their study showed that unexpected changes in the trajectory of COVID-19 infections predict US stock returns in real time. Their study further revealed that unanticipated doubling of projected infections forecast a next-day decrease in aggregate US market value of 4 to 11 percent, indicating equity markets will rebound even in the persistence of the pandemic. Haacker [52] investigated the economic cost of the HIV/AIDS pandemic. His study revealed that HIV has caused financial losses to affected families and torn families apart. Studies by Bloom et al. [53] warned the world about the possible emergence of a pandemic. They stressed that the world was not ready now to accommodate a pandemic and suggested buffering our economic indicators to survive in the event of a pandemic. The present occurrence of COVID-19 pandemic suggests that the world has not paid heed to this warning. Dreyer et al. [54] investigated the frequency of influenza (flu) [2] pandemics on the South African insurance industry and studied the effects it had on the industry. Their study concluded that a mild pandemic will cost the South African Economy R 1.1 billion claims excluding annuity, while a severe pandemic could cost R 55 billion claims.

At present, insurers in Ghana are counting their losses as a result of COVID-19, but the pandemic could also present an opportunity for a stronger bounce back. The most obvious effect of COVID-19 on the insurance industry in Ghana is the upsurge in health, travel and business claims. Though insurance policies do not directly cover pandemics, the impact of COVID-19 on the global economy has had a toll on the insurance industry. There are reports of insurers already paying millions of cedis through e-payment channels even during the lockdown period. There is obvious pressure on sales from reduced business activity and less use of face-to-face channels in transactions. Though the lockdown in Ghana was brief, it further exacerbated the impact of the pandemic resulting in lowering interest rates and increasing credit risk. The National Insurance Commission of Ghana (NIC) reported that many more clients now call in to complain about their insurers, and the number of people calling in to enquire about claims has quadrupled within the pandemic period as compared to the same period last year. Most aviation passengers insist that insurers should refund the premiums they paid since there is a ban on travel. Business owners all over the country have also called in to demand claims for interruption of businesses. The NIC indicates that it is currently studying the impact of the pandemic and will soon come out with guidelines and new policies on how to survive. At present, there is no study to investigate the impact of COVID-19 on the insurance industry in Ghana. Insurers either cannot simulate impacts of the pandemic on their companies or are currently overwhelmed by mounting operational pressures to be able to study the impacts of the pandemic. This study aims to investigate the impact of COVID-19 on the insurance industry in Ghana and discuss solutions as well as project future expectations. We are covering the impact of the pandemic from the month of the first reported case count until the present moment of this study (i.e., March to June 2020). The specific objectives of the study are:(i)to estimate the impact of COVID-19 on the insurance industry in Ghana.(ii)to discuss insurer expectations and present solutions.

## 2. Materials and Methods

### 2.1. Sources of Data

Data for this study include operations data and documents from insurance companies, Ghana Health Service (GHS), World Health Organization (WHO), Ghana Statistical Service Department (GSS), Ghana National Insurance Commission (NIC), Ghana Labour Commission (LC) and the Ghana National Health Insurance Authority (NHIA). The Data includes population data which comprises, gender specification, job, insurance status, type of insurance, income and other social or economic indices. Data from Ghana Health Service and WHO consist of updates of COVID-19 case counts, infection rates, deaths and total recoveries, as well as information about responses to the pandemic. From the insurance companies, the data consist of total premiums, claims, profits, assets and liabilities, market share and business investments. Insurance data was collected as far back as 2013 to compare trends of operations.

### 2.2. Data Collection

There are presently 141 regulated insurance companies in Ghana comprising 24 life insurance companies, 29 non-life insurance companies, 3 reinsurance companies and 85 insurance brokers with more than GH Ȼ6 billion total assets.

To measure the impact of COVID-19 on the insurance industry, we designed and administered three different types of questionnaires to three types of responders. The first category of responders includes statutory actuaries. The questionnaire was designed to capture the level of risk the pandemic posed on insurance companies; responses of insurers; actions and possible solutions; expectations of insurers and their preparation. Category 2 responders include insurance officers in the category of general manager, operations manager, supervisors, public relations officers, human resources officers, logistic officers, accountants and Liaison officers. Questions for this category investigate the financial losses to insurance companies and current relations with clients. The last category of people were clients, policyholders, stakeholders and partners of insurance companies as well as the general public who are stakeholders of the insurance industry. This category of questions investigates customer’s trust in the insurance company as well as how they were treated by the insurance companies before and during the pandemic. Details of the questionnaire are presented in Appendix A
as document Q1. Category 1 and 2 questionnaires were sent as official letters to the responders while category 3 was administered via social media platforms including Ghanaweb.com, Facebook.com, Twitter, Jobsinghana.com and on WhatsApp messenger. In category 1, 5 actuaries were contacted out of a total of 7 qualified professional actuaries in Ghana. In category 2, 200 letters were sent via email and WhatsApp to responders. The number of responders interviewed in category 3 consists of 50 people on face-to-face interviews, 400 via WhatsApp messenger, 656 via Facebook and 468 on Ghanaweb.com. In total, 1779 responders from the 16 administrative regions in Ghana were interviewed.

## 3. Results

### 3.1. General Outlook of the Ghanaian Insurance Industry

The outlook of the Ghanaian insurance industry since the pandemic started from March until June shows an economic recession. There are indications of reducing profits and premiums, stable assets and market share but increasing claims. Figure 1 compares the trend of total profits, premiums, claims and assets for the quarterly period March to June from 2013 to 2020. Insurance accounting on excel spreadsheet can be found in Appendix A as Appendix A.

### 3.2. Customer Complaints

The results present customer complaints in the insurance industry with companies categorized into life and non-life. The trend of customer complaints in the industry from March to June is compared from 2014 to 2020. For life insurance companies, the trend indicates an increase in the number of complaints until 2016. The number of complaints then decreased down to 2017. There was a sharp increase in complaints from 2017 to 2018 and then a decline. Customer complaints in non-life insurance companies were similar to life insurance companies. Figure 2 presents a comparison of customer complaints on life and non-life insurance companies.

### 3.3. Impact on Life and Non-Life Insurance Policies

We have, in this section, compared life and non-life insurance companies to see the impact on human life and health. The results show that premiums of all 28 non-life insurance companies dropped in 2020 and the same for life insurance companies. Claims, however, increased for both life and non-life insurance companies within the period. Figure 3 compares performance of life and non-life insurance companies. 

### 3.4. The Impact on Health Insurance Policy: The Role of the Ghana National Health Insurance Scheme

Broadly, health insurance has experienced some of the operational challenges general insurance and life insurance faced. Ghana has a National Health Insurance Scheme (NHIS), one of the few in Sub-Saharan Africa. The National Health Insurance Authority (NHIA) superintends the operation of the national health insurance policy that works to ensure access to basic health care for all Ghanaians. The NHIA also licenses and regulates District-level Mutual Health Insurance Schemes (DMHISs) and accredits others. Currently, there are 145 district health insurance schemes including 10 in Greater Accra, the capital of Ghana. The national health insurance does not provide coverage for COVID-19, however, as a government institution, the NHIS partnered with the government to provide logistics and PPEs to fight the pandemic. Under the National Health Insurance ACT, 2012 (ACT 852), Section 40 sets aside a National Health Insurance Fund (NHIF) that qualifies to support the nation in pandemic situations. Section 40 (2 c and d) states that the purpose of the NHIF is to facilitate provisions of health facilities or invest in programs that promote the good health of Ghanaians. In this regard, the NHIA supported the government efforts in fighting COVID-19 by donating GH Ȼ250,000 to the COVID-19 relief fund set-up by the government. The NHIA also donated several Personal Protective Equipment (PPEs), food and other items to communities, churches, schools and mosques at various municipal and district levels. Private Health Insurance providers have also witnessed increased spending on their Corporate Social Responsibility (CSR) budgets to support communities with food and PPEs and to rebate clients who were facing financial challenges during the partial lockdown.

### 3.5. Impact on Travel Insurance

COVID-19 has brought the travel industry especially aviation to a standstill. As the pandemic intensified, there was a ban on travel and closure of airports in Ghana. Ghana presently has the second busiest airport in West Africa with an average of 56 flights a day. Based on developments so far (quarantine measures, border closures, travel bans) and compared with patterns of similar pandemics (2003 SARS and 2009 global economic crisis), we estimate the impact of the pandemic on the growth of the Ghanaian travel industry to −32% on Revenue Passenger Kilometres (RPK) and growth of −4% on passenger revenue. During the partial lockdown, travel insurers stopped sales since there were no travelers. There were also more refunds due to cancellation of travel policy as a result of unutilized travel days as the insured cannot use their annual travel plans. Some insurers have started laying off workers and switching full-time employees to part-time work. Insurers must consider the purpose for which travel insurance was bought and therefore protect the interest of the customer. They must also protect their commercial interest as loss cannot guarantee quality service or keep customers happy. As a result of a reduction in travel, the auto insurance industry is subscribing partial premium credits for customers to reduce claims.

### 3.6. Virtual Workforce

We discovered that some companies had to perform most of their operations virtually due to social distancing guidelines and other government restrictions. This has created business continuity challenges for companies, employers and employees. It has particularly affected actuaries who handle large data sets models that require significant computing power and capacity. Having to remotely manage these operations adds additional complexity and difficulty in systems access and processing times. The abrupt and unplanned transition to a remote working environment has forced increased collaboration between actuarial departments, finance and IT to meet targets.

### 3.7. Level of Preparedness, Expectations and Hopes of Ghanaian Insurers

Ghanaian insurers and reinsurers including actuaries were asked about their level of preparedness for a pandemic. They were particularly asked to assess the level of threat the pandemic poses to insurance companies, the measures insurers have in place to protect their companies, whether insurers have attempted to quantify the risk and the extent to which profit and solvency are threatened by COVID-19. Their responses are presented below:In general, insurers were optimistic that their assets cover regulatory minimums sufficient to allow for a pandemic such as COVID-19, but 25% were doubtful whether if COVD-19 should persist beyond 12 months, they will survive insolvency.Most insurers have attempted to model the risk of the pandemic closely while others tried modeling their results under stress testing, but the majority have done nothing. This raises a big concern and needs to be considered by the National Health Insurance Commission. Elsewhere in the world, it is beneficial for National Health Insurance Authorities to implement readiness audits of all insurance companies to assess their finances and assets and model the impact future pandemic will have on them.All companies agreed that COVID-19 has had an operational economic impact on them, and they also agreed that they were not 100% prepared for the pandemic. They were however optimistic of surviving the pandemic and hopeful of a successful bounce back.Companies fear that existing products with guarantee rates are likely going to be unable to recoup losses as a result of COVID-19. They believe that even where rates are annually reviewable, it will probably not be feasible to recover those losses in future premiums.Regarding new products, little has been done to anticipate the impact COVID-19 will cause.All the insurers were of the view that annuity savings will protect some of their assets, but many insurance companies had little or no annuity businesses.

Summary of insurer responses to questionnaires are presented in Appendix A in Appendix A.

## 4. Discussion

### 4.1. The General Outlook

Due to reduced economic activity, premiums have reduced and are expected to reduce further until the end of the year. Profits have been significantly reduced because more claims are being paid out than premiums collected over the period. Some companies had to lay off employees thus reducing productivity. Budgets of companies have also increased due to more spending on social responsibility to help the government fight the pandemic. For instance, some insurers had to buy hand sanitizers, gloves, nose masks and other PPEs for their workers. Food and other provisions were also bought for communities during the three-week lockdown. Due to the volatility in the financial markets, investment income has significantly dropped low with reduced interest rates. Insurers may also experience an increase in policy lapses in certain segments of their business if individuals are unable or choose not to pay premiums to keep their policies active. This trend will continue until the end of 2020. Companies that cannot keep track of failed policyholders who may want to repurchase insurance are likely to have an advantage over competitors in regenerating business. Insurers and actuaries are advised to monitor the trends carefully to inform changes in the processes.

Claims have drastically increased along many lines. As companies transition to virtual working environments, most companies have not seen the impact on claim processing. Companies are advised to carefully evaluate the volume of claims received both during the pandemic and as it subsides, as well as their efficiency to process claims virtually. Loss of lives and unproductivity have resulted in some insurers having to rebate their clients to survive during the lockdown period. This may likely extend till the end of 2020. Event cancellation has caused a huge loss to the insurance industry. The Ghana premier league is currently on a standstill. Major music and comedy concerts are postponed, and other sporting events have been put on hold. During the lockdown, all large public gatherings for music, worship and celebration were canceled and will surely struggle to get insurance coverage in the future. Travel insurance has become more expensive since the WHO declared the disease a pandemic.

Agriculture and oil insurance is expected to receive more payment of claims as Ghana is an agricultural country. Dropping cocoa, forest products, oil and gold prices have further increased claims. Oil insurance is a major contributor to the Ghanaian insurance industry since 2017.

Claims are also increasing in the short term due to supply-chain disruption and loss of revenues. Liability insurance claims can be demanded where contraction of COVID-19 or death as a result of it is attributed to negligence. For example, clients can litigate hotel owners for not taking precautionary measures to prevent the virus or for not operating an effective quarantine within their premises. There could also be lawsuits where the virus is brought into hospitals and nursing homes and causes fatalities. Presently, private schools in Ghana are afraid to operate for fear of litigation should a pupil be infected. Churches and other prayer centers are afraid to open even though the lockdown has been lifted by the president. Workers could demand compensation pay-outs where employees have been infected at work especially commercial taxi and bus drivers driving cars of commercial transport operators, aircraft flight crew workers, hospital workers and general company workers.

Customer complaints have increased because liquidity is beginning to hit some insurance companies, and as a result, payment of claims has become challenging. Secondly, some clients do not understand that insurance policies do not cover pandemics. Insurers of life and health policies might not have properly explained to their clients how insurances do not cover pandemics, and since COVID-19 is a novel virus, customers and insurers will disagree as to where to accurately place the pandemic within the insured and non-insured diseases category of health insurance. This has caused lots of arguments between insurers and their clients thereby increasing customer complaints to the NIC. On business interruption claims, some insurers are arguing that the interruption on business was brief and does not qualify for claims. On travel insurance, some customers complained that recovering losses especially from airlines and travel agencies has been difficult as insurers have challenged validating claims.

### 4.2. How Insurers Should Respond to the Crises

The Ghanaian insurance industry is responding to the COVID-19 on multiple fronts as capital managers, employers, and claim payers. Each insurance company has its distinct challenges; however, the most pressing concern for all insurers is protecting the health and safety of their employees and partners including brokers and agents in the insurance community. The focus of insurers should be to review and update their crisis management plans and take steps to continue operations with little disruption to clients. Insurers should consider establishing temporal multi-purpose emergency decision-making units purposely for this pandemic to coordinate responses and set new safety protocols to contain shocks. They should also set up a comprehensive communications system to keep employees, clients, distributors and other partners fully informed about the status of business continuity plans and information on how to remain safe.

Insurers should also adapt to working from remote locations. They should enable and equip company staff from actuaries, underwriters and claim managers to work offsite, most probably from homes. They should particularly make it possible for employees to access necessary files from homes. Furthermore, chief information security officers should be equipped to establish new cybersecurity protocols to ensure the safe exchange of highly classified information among employees connecting from homes. Insurers should particularly make sure that chief technology officers, chief information officers, and chief information security officers have the following technological capabilities: (a) a virtual private network to privately connect remotely for critical business operations, (b) a personal laptop or desktop computer issued by the company, (c) video conferencing tools like Zoom and Tencent meeting with training on how to use them, and (d) a competent, well-equipped IT support team to answer employees questions and help employees from offsite.

Insurers should plan training and equipping staff to work under social distancing regulations since there is potentially a long time to go for the pandemic. With good digital tools, this should not be a problem for insurers. This could potentially be a period of productive planning, training and outreach to stakeholder groups at remote distances. Lastly, insurers should establish a risk management team to assess how quickly and effectively they were able to respond to crises in the pandemic period. They should also determine any additional steps that may need to be taken to adapt to their organizations and make them more resilient if faced with future pandemics like the type of COVID-19.

### 4.3. Managing and Quantifying Loss During a Pandemic

Quantifying and managing loss are very critical to insurers during a pandemic. If a claim is presented, it will be complex to measure the loss. Insurers can adopt the following procedures to measure loss:By acting promptly: Early engagement with the client is very important to win the trust and understand the potential impacts of the pandemic.Understand the drivers of business specific to the insured’s business model. The insured must not be in say Accra, Kumasi or Tema (high COVID-19 case count cities in Ghana) to suffer a potential business interruption loss. Businesses operate within a long supply chain, and therefore, contingent business interruption tends to occur in the weaker part of the supply chain, which could originate from any part of the country and can lead to large losses in business.Keep an accurate trace of cause and effect. As losses accrue, the ability to keep an accurately documented trail to prove the direct causal link between the insured peril and financial losses is very critical.Keep track of worldwide activities and business trends: It is very prudent to consider the financial impact of the pandemic on other areas of the world.Seek early advice from professionals. Pandemics such as SARS and H1N1 have impacted insurance industries in other parts of the world, and insurance professionals have experience operating in pandemic situations. The early seeking of advice may save losses.

### 4.4. Other Solutions and Actions for Insurers

COVID-19 occurrence has exposed the weaknesses in the Ghanaian economy and insurance industry. We have now seen how the country’s economy buckles under a major threat like a pandemic. The questions that need to be asked are the following: how can insurers prepare for a pandemic? What issues need to be considered to reduce the impact of a pandemic in its next occurrence, and what questions need to be addressed during the recovery process? The following points address issues that need to be considered to reduce the impact of a pandemic on mortality, morbidity and social disruption:Giving up-to-date and truthful information to parties: Most often, due to politics, governments and agencies try to hide the true scale of a pandemic by under-reporting on infection rates, case counts and deaths.Getting prepared ahead of time: There are more worries about the likelihood of a pandemic occurring today than there were 40 years ago. WHO and other partners have consistently warned the general public of the likely occurrence of a pandemic. This should, therefore, give insurers and hospitals enough reason to prepare adequately for a pandemic. The more time and resources spend preparing, the more effective a global response shall be. Vaccine development is a very important part of preparation. The government should maintain and support infectious disease control and prevention units such as the University of Ghana medical school, the Kwame Nkrumah University of Science and Technology School of Medical Sciences, the University for Development Studies School of Medical Sciences, the Noguchi Medical Research Institute, the Kumasi Centre for Collaborative Research In Tropical Medicine and some nursing training colleges such as the Jirapa Nursing and Midwifery Training College to be able to do research into virology and build the capacity of health workers. The aim is to promote research into developing vaccines that will be stocked in readiness for a pandemic. Antivirals should also be prepared in waiting for a pandemic situation.

### 4.5. Recovering from the Pandemic

To recover from a pandemic will take the unified collaboration of several stakeholders including the Ministry of Health, Insurance Industry, Banking Sector, Police Department and Customs Excise and Preventive Service. Global recovery depends on the preparedness of the international community as a whole and the severity of the pandemic. More important is the preparedness of the insurance industry, which must work extra hard during the pandemic period. Total recovery will only take place in a post-pandemic environment, and companies who were not well prepared will suffer the consequences of their unpreparedness.

### 4.6. What the World Bank Must Do to Help Client Countries

The World Bank should assist insurance supervisory agencies to help them survive and recover. The insurance industry will need assistance in crisis response strategies, training and facilitation to act decisively and help in upgrading their communication capacity. The World Bank can also provide support by providing solvency and capital and asset-liability management, mentoring training and assisting supervisors to act in a more complex environment.

## 5. Conclusions

The COVID-19 pandemic has posed a sudden and unexpected shock to the insurance industry. The financial impact is huge with profits dropping by 16.6% within the period under review—March to June 2020. Total premiums have dropped by 17.01% while claims have increased by 38.4%. Most companies have reduced market share with only a few maintaining their market share. The estimated financial loss to the Ghanaian insurance industry within the period under review is GH Ȼ112 million.

Insurance companies were affected differently depending on different factors such as liquidity, their portfolio at risk, reliance on reinsurance, level of free assets and protection that reinsurers have in place.

The initial response of insurers was poor as the Ghanaian insurance industry does not have sufficient experience in managing a pandemic situation, but recovery was quick as most insurance companies have now adapted to working from remote locations and enhanced their IT as well as security protocols. Insurance managers have also responded by issuing statements and taken specific actions to calm down panic reactions among policyholders. They have particularly simplified the claim process to make policyholders access claims easily and set new rules to regulate the sector. Insurers have also increased measures for underwriting claims service. The government has set up a COVID-19 fund to fight the pandemic. There is also a fund for frontline health workers to be compensated for risking their lives.

Though the loss in finance is expected to continue until the end of the year, the forecast shows a bounce back in operations that will resume normal accounting on claims, premiums and profits in early 2021. This growth is expected to be sustained for five years, but insurers must adapt to the new mode of business transaction and be able to quantify and manage their losses.

The World Bank can step in to assist insurers with training on crisis response strategies, training and facilitation to act decisively and offer financial bail-out to insurers who are unable to recover.

While the pandemic persists, insurers must learn to give up-to-date and truthful information to their clients; they must be proactive rather than reactive in their response and be prepared at all times for worst-case scenarios. They must continue to relate with clients in a positive working environment to maintain their trust.

## Figures and Tables

**Figure 1 ijerph-17-05766-f001:**
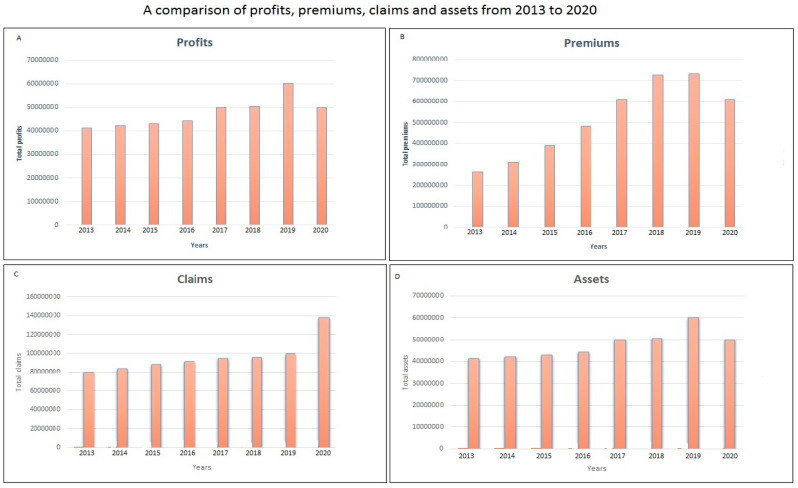
General quarterly (March to June) outlook of the insurance industry: (**A**) trend of profits, (**B**) trend of premiums, (**C**) trend of claims and (**D**) trend of assets.

**Figure 2 ijerph-17-05766-f002:**
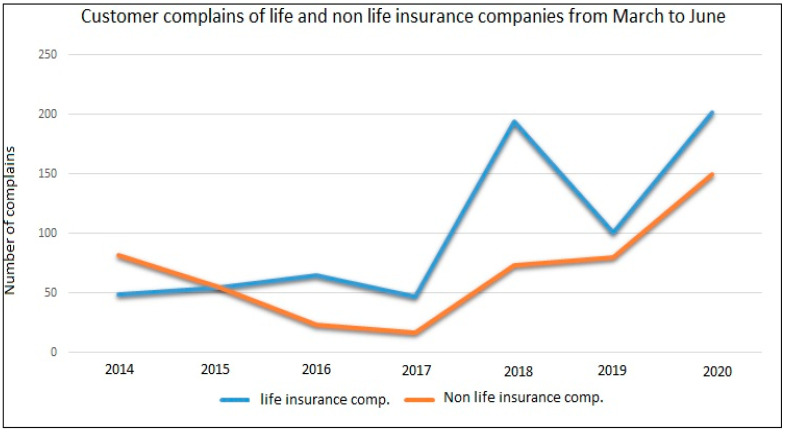
Comparison of customer complaints in the insurance industry within the period under review.

**Figure 3 ijerph-17-05766-f003:**
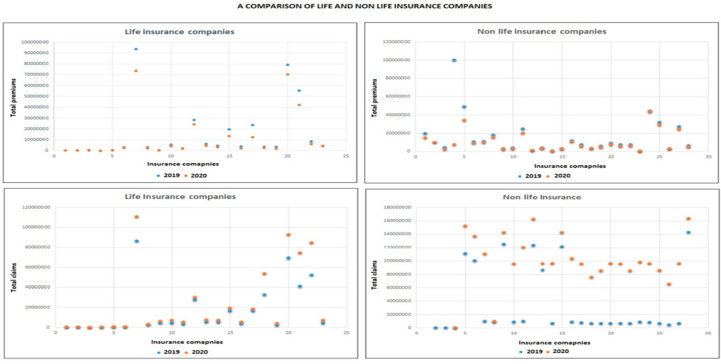
Life and non-life insurance companies’ performance compared.

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
