# Peer review of "The Impact of COVID-19 on the Insurance Industry"

_ijerph, 2020, doi:10.3390/ijerph17165766_

Round 1

Reviewer 1 Report

The paper intends to investigate and evaluate the impact of COVID-19 on the insurance industry in Ghana from March to June 2020.  Qualitative and quantitative methodologies are employed to discuss the impacts and forecast future trends.

The topic is eye-catching; however, I have some major concerns about the validity of the study in terms of data, literature review, the model, methodologies, and conclusions drawn for the research framework. Comments are as follows:

Main Comments

  1. Data

There seem to be two types of sources in terms of data: statistics form insurance companies, Ghana government, and organizations, and WHO; and surveys from three types of responders. Quantitative analysis is based on the statistics and qualitative discussions regarding responses of insurers and actions with possible solutions and some expectations are built upon the surveys/questionnaires. The two-part studies should be inherently related, but not well connected in the article.

  1. Literature review

This article has a comprehensive collection of facts and estimates regarding the previous epidemic losses. However, the major literature of economics/financial analysis regarding disaster/epidemic is missing, which directly results in the weakness of model selections.

  1. Model of forecasting

I don’t think the evaluation of forecasting methods are fairly or accurately stated in the “Introduction” (Lines 126-147, Page 3). I am especially concerned about the remarks like “…do not give/lack a detailed analysis”, which is confusing to explain the scenarios for a specific method to be applied to. To my best knowledge, without controlling for confounding factors, simple comparisons between two years' trend are not reliable to draw conclusions on the impact of the pandemic. Moreover, Simple Exponential Smoothing is not appropriate for forecasting economic/financial indicators with a major event. DID or Response Functions should be employed. Even for the SESM, the model specification is very confusing, some notations are not explained (e.g., qi). How the parameters, e.g., weights, are selected are not explained either. No tests have been run to test the robustness of the model, not even root mean squared errors (RMSEs).

I believe a much more comprehensive and selective literature search would be helpful, here are some papers I found that would be helpful, but there are much more out there, some can be found in their references:

Barro, R. J., Ursúa, J. F., & Weng, J. (2020). The coronavirus and the great influenza pandemic: Lessons from the “spanish flu” for the coronavirus’s potential effects on mortality and economic activity (No. w26866). National Bureau of Economic Research.

Cerra, V., & Saxena, S. C. (2008). Growth dynamics: the myth of economic recovery. American Economic Review98(1), 439-57.

Alfaro, L., Chari, A., Greenland, A. N., & Schott, P. K. (2020). Aggregate and firm-level stock returns during pandemics, in real-time (No. w26950). National Bureau of Economic Research.

  1. Moreover, the paper claims to forecast for two scenarios: (1) COID-19 ends shortly or (2) COVID-19 prolongs. I couldn’t locate this section in the paper.
  2. Survey data need to be summarized in Tables. Response rate, the ratio of different responses all need to be reported. I was not able to open the questionnaire for some reason. Pardon me if tables are included in that .rar file.

I believe the paper can be greatly improved with the data available. All it needs is a more rigorous model and appropriate treatment.

Author Response

Dear reviewer,

Thank you for your effort in pointing out weaknesses in parts of our manuscript. We have carefully considered all your comments and corrected them step-by-step in the manuscript as well as we can.

The changes we made are highlighted in red in the manuscript. We look forward to hearing from you soon and hope that our manuscript will finally get accepted for publication. 

Find attached full reponses to your comments

Reviewer 2 Report

The article deals with a topic of great interest, such as the effect of COVID-19 in the insurance industry. I think the authors have done a good job of handling the data, and that a detailed analysis of different aspects of the sector is presented.

However, I consider that there are some aspects that should be improved so that the article could be considered for publication.

Firstly, the structure of the paper is confusing, making it difficult for the reader to follow the steps taken by the authors. The paper needs a theoretical section where the authors place their own study in the context of other studies. Based on this literature review they should identify a gap or problem that the study will address. In addition, based on that revision they should propose their research hypotheses, with a clear connection with the review of the literature (these hypotheses could be in the paper implicit, but clear to the reader). From my point of view many of these aspects are assumed in the paper, but I think the authors should make an effort to improve the way they present it. I think the best way to do this is to separate the introduction, in which the paper is presented, from that new section that includes the theoretical support of the research.

Second, page 3 lists the different time series analysis methods that can be used, and ultimately decides on the simple exponential smoothing technique. I think that all this part should be located in the methodology section, and the use of this technique, which also has its drawbacks, should be more justified. In addition, I think that the limitations of the article should be those that derive from the use of time series analysis to make projections in a totally new situation such as COVID-19.

Finally, I think that in the exploitation of the information from the questionnaire, some quantitative indicators should be used to analyze the relevance of the results presented. Although it is a more qualitative part, tests can be carried out to analyze the significance of the answers. At least a table with the descriptive statistics of the survey and the questions analyzed should be included, so that the reader can have a more faithful image of the work presented.

I think that with these to improvements the paper would be suitable for publication, the remain of the study is well executed. 

Author Response

Dear reviewer,

Thank you for your hard work to point out weaknesses in our paper. We have considered your comments step by step and responded to them all in the attached document

Reviewer 3 Report

Dear Author/s,

Thank you very much that I could read and evaluate this manuscript and I hope that my comments could improve the quality of this interesting research.

This article presents a very significant scientific problem and is extremely interesting for potential readers in my opinion.

The study contributes to filling the literature gap by describing the Impact of COVID-19 on the Insurance Industry in Ghana. I will not indicate the Ghana case in the title of the article, not to limit the number of potential readers to this location. This area of research could be pointed out in methodology part.

The weak point of this study is the methodology and prediction part. I have not found the presentation of predictions taking into account the time span scenarios: 1. Covid-19 will last in short term and 2. COVID-19 will last in the long term run.

Introduction tells the prospective readers what they can expect.

In my opinion the article includes too many keywords, namely: (…Claims; Coronavirus; COVID-19; Insurance industry; Infection rate; National Insurance Commission; Pandemic; Premium; Social distancing; World Health Organization…) Please limit keywords number to the most important ones.

I will write the sentence below as follows (please compare):

"...Coronavirus, scientifically reclassified as COVID-19 assumed global pandemic proportions since its inception in December 2019 from Wuhan, China [1]...."

Coronavirus disease, scientifically reclassified as COVID-19 assumed global pandemic proportions since its inception in December 2019 from Wuhan, China [1].

Could you please consider not indicate when SARS-CoV-2 starts spreading in the sentence above, because some scientists claims that it was even earlier then December 2019.

„…injection of US $ 1 billion…” - Please change the term „injection” in the context of financial aid or financial support (please use professional financial vocabulary here).

I will add the sources of figures below them.

Author/s use/s sufficient number of literature sources (57 positions).

Author Response

Dear Reviewer,

Thank you very much for all your hard work to review our paper. We have considered all your comments and responded to them step by step in the attached document.

Round 2

Reviewer 1 Report

I have read carefully the response and the changes made to the manuscript. I highly appreciate the drastic efforts devoted and attention paid to my comments. However, with the obviously improved manuscript, I still feel that my major concerns have not been fully addressed, especially regarding the literature review and the econometric methodology.

For the literature review, it is not the quantity that matters, but the relevance. I provided three references in my previous review comments which I felt more important and relevant to the study. However, I think none of them was considered. 

For the econometric methodology, I think I emphasized that Holt-Winters Exponential Smoothing Method (in the manuscript, the "s" of "Winters" is missing) is not a good technique when there is a major change in the data structure, which is a common weakness of time series analysis without econometric modeling, and DID analysis is a much better candidate. The kink in the figures just represented a weighted average of forecasted value in the past period and dropped current value due to the pandemic, sure, with adjustments of seasonality and trend, etc. However, the assumption is the trend, seasonality will continue as the pre-pandemic period after this sudden drop. Given the changes analyzed in the study, I would not take it as a reasonable assumption. Moreover, I don't think the author(s) justified their choice appropriately. All the papers cited are just applications of this method in financial series, not for similar impact studies. It should be noted that a method suitable for the data and purpose of the study is supposed to be identified. The Holt-Winters Exponential Smoothing Method, great for a lot of financial studies, is still a bad choice for this proposed study, which impaired the projections, totally.

I actually appreciate the data and analysis and actually think taking out the time series wholly while keeping the survey and analysis could be a way around. The two parts are not necessarily attached to each other.

Some minor comments are related to the responses to the reviewers. I cannot see the responses to other reviewers' comments, however, in my case, not every comment was carefully responded. Also, please try to avoid using "the seeming mention of 2019 and 2020 in some parts of the manuscript is to highlight..." This is not professional. 

Author Response

Dear Editor, please find attached, our point by point response to your comments.
